# Diffusion Models for Causal Discovery via Topological Ordering

**Pedro Sanchez**[1][*] **, Xiao Liu**[1]**, Alison Q O'Neil**[2,1]**, Sotirios A. Tsaftaris**[1,3]
[1]The University of Edinburgh
[2]Canon Medical Research Europe
[3]The Alan Turing Institute

## Abstract

Discovering causal relations from observational data becomes possible with additional assumptions such as considering the functional relations to be constrained as nonlinear with additive noise (ANM). Even with strong assumptions, causal discovery involves an expensive search problem over the space of directed acyclic graphs (DAGs). *Topological ordering* approaches reduce the optimisation space of causal discovery by searching over a permutation rather than graph space. For ANMs, the *Hessian* of the data log-likelihood can be used for finding leaf nodes in a causal graph, allowing its topological ordering. However, existing computational methods for obtaining the Hessian still do not scale as the number of variables and the number of samples are increased. Therefore, inspired by recent innovations in diffusion probabilistic models (DPMs), we propose *DiffAN*[1], a topological ordering algorithm that leverages DPMs for learning a Hessian function. We introduce theory for updating the learned Hessian without re-training the neural network, and we show that computing with a subset of samples gives an accurate approximation of the ordering, which allows scaling to datasets with more samples and variables. We show empirically that our method scales exceptionally well to datasets with up to $500$ nodes and up to $10^5$ samples while still performing on par over small datasets with state-of-the-art causal discovery methods.

## 1 Introduction

Understanding the causal structure of a problem is important for areas such as economics, biology (Sachs et al., 2005) and healthcare (Sanchez et al., 2022), especially when reasoning about the effect of interventions. When interventional data from randomised trials are not available, causal discovery methods (Glymour et al., 2019) may be employed to discover the causal structure of a problem solely from observational data. Causal structure is typically modelled as a directed acyclic graph (DAG) $\mathcal{G}$ in which each node is associated with a random variable and each edge represents a causal mechanism i.e. how one variable influences another.

However, learning such a model from data is NP-hard (Chickering, 1996). Traditional methods search the DAG space

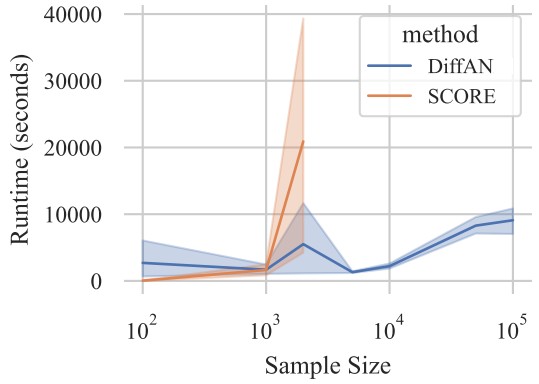

Figure 1: Plot showing run time in seconds for different sample sizes, for discovery of causal graphs with 500 nodes. Most causal discovery methods have prohibitive run time and memory cost for datasets with many samples; the previous state-of-the-art SCORE algorithm (Rolland et al., 2022) which is included in this graph cannot be computed beyond 2000 samples in a machine with 64GB of RAM. By contrast, our method DiffAN has a reasonable run time even for numbers of samples two orders of magnitude larger than capable by most existing methods.

---

[*]pedro.sanchez@ed.ac.uk

[1]Implementation is available at `https://github.com/vios-s/DiffAN`.

by testing for conditional independence between variables (Spirtes et al., 1993) or by optimising some goodness of fit measure (Chickering, 2002). Unfortunately, solving the search problem with a greedy combinatorial optimisation method can be expensive and does not scale to high-dimensional problems.

In line with previous work (Teyssier & Koller, 2005; Park & Klabjan, 2017; Bühlmann et al., 2014; Solus et al., 2021; Wang et al., 2021; Rolland et al., 2022), we can speed up the combinatorial search problem over the space of DAGs by rephrasing it as a *topological ordering* task, ordering from leaf nodes to root nodes. The search space over DAGs with $d$ nodes and $(d^2 - d)/2$ possible edges is much larger than the space of permutations over $d$ variables. Once a topological ordering of the nodes is found, the potential causal relations between later (cause) and earlier (effect) nodes can be pruned with a feature selection algorithm (e.g. Bühlmann et al. (2014)) to yield a graph which is naturally directed and acyclic without further optimisation.

Recently, Rolland et al. (2022) proposed the SCORE algorithm for topological ordering. SCORE uses the Hessian of the data log-likelihood, $\nabla_x^2 \log p(\mathbf{x})$. By verifying which elements of $\nabla_x^2 \log p(\mathbf{x})$'s diagonal are constant across all data points, leaf nodes can be iteratively identified and removed. Rolland et al. (2022) estimate the Hessian point-wise with a second-order Stein gradient estimator (Li & Turner, 2018) over a radial basis function (RBF) kernel. However, point-wise estimation with kernels scales poorly to datasets with large number of samples $n$ because it requires inverting a $n \times n$ kernel matrix.

Here, we enable *scalable* causal discovery by utilising neural networks (NNs) trained with denoising diffusion instead of Rolland et al.'s kernel-based estimation. We use the ordering procedure, based on Rolland et al. (2022), which requires re-computing the score's Jacobian at each iteration. Training NNs at each iteration would not be feasible. Therefore, we derive a theoretical analysis that allows updating the learned score without re-training. In addition, the NN is trained over the entire dataset ($n$ samples) but only a subsample is used for finding leaf nodes. Thus, once the score model is learned, we can use it to order the graph with constant complexity on $n$, enabling causal discovery for large datasets in high-dimensional settings. Interestingly, our algorithm does not require architectural constraints on the neural network, as in previous causal discovery methods based on neural networks (Lachapelle et al., 2020; Zheng et al., 2020; Yu et al., 2019; Ng et al., 2022). Our training procedure does not learn the causal mechanism directly, but the score of the data distribution.

**Contributions.** In summary, we propose DiffAN, an identifiable algorithm leveraging a diffusion probabilistic model for topological ordering that enables causal discovery assuming an additive noise model: (i) To the best of our knowledge, we present the first causal discovery algorithm based on denoising diffusion training which allows *scaling* to datasets with up to 500 variables and $10^5$ samples. The score estimated with the diffusion model is used to find and remove leaf nodes iteratively; (ii) We estimate the second-order derivatives (score's Jacobian or Hessian) of a data distribution using neural networks with diffusion training via backpropagation; (iii) The proposed *deciduous score* (Section 3) allows efficient causal discovery *without* re-training the score model at each iteration. When a leaf node is removed, the score of the new distribution can be estimated from the original score (before leaf removal) and its Jacobian.

## 2 PRELIMINARIES

### 2.1 PROBLEM DEFINITION

We consider the problem of discovering the causal structure between $d$ variables, given a probability distribution $p(\mathbf{x})$ from which a $d$-dimensional random vector $\mathbf{x} = (x_1, \ldots, x_d)$ can be sampled. We assume that the true causal structure is described by a DAG $\mathcal{G}$ containing $d$ nodes. Each node represents a random variable $x_i$ and edges represent the presence of causal relations between them. In other words, we can say that $\mathcal{G}$ defines a structural causal model (SCM) consisting of a collection of assignments $x_i := f_i(Pa(x_i), \epsilon_i)$, where $Pa(x_i)$ are the parents of $x_i$ in $\mathcal{G}$, and $\epsilon_i$ is a noise term independent of $x_i$, also called exogenous noise. $\epsilon_i$ are i.i.d. from a smooth distribution $p^\epsilon$. The SCM entails a unique distribution $p(\mathbf{x}) = \prod_{i=1}^{d} p(x_i \mid Pa(x_i))$ over the variables $\mathbf{x}$ (Peters et al., 2017). The observational input data are $\boldsymbol{X} \in \mathbb{R}^{n \times d}$, where $n$ is number of samples. The target output is an adjacency matrix $\boldsymbol{A} \in \mathbb{R}^{d \times d}$.

The **topological ordering** (also called *causal ordering* or *causal list*) of a DAG $\mathcal{G}$ is defined as a non-unique permutation $\pi$ of $d$ nodes such that a given node always appears first in the list than its descendants. Formally, $\pi_i < \pi_j \iff j \in De_{\mathcal{G}}(\mathrm{x}_i)$ where $De_{\mathcal{G}}(\mathrm{x}_i)$ are the descendants of the $ith$ node in $\mathcal{G}$ (Appendix B in Peters et al. (2017)).

## 2.2 Nonlinear Additive Noise Models

Learning a unique $\boldsymbol{A}$ from $\boldsymbol{X}$ with observational data requires additional assumptions. A common class of methods called additive noise models (ANM) (Shimizu et al., 2006; Hoyer et al., 2008; Peters et al., 2014; Bühlmann et al., 2014) explores asymmetries in the data by imposing functional assumptions on the data generation process. In most cases, they assume that assignments take the form $\mathrm{x}_i := f_i(Pa(\mathrm{x}_i)) + \epsilon_i$ with $\epsilon_i \sim p^\epsilon$. Here we focus on the case described by Peters et al. (2014) where $f_i$ is nonlinear. We use the notation $f_i$ for $f_i(Pa(\mathrm{x}_i))$ because the arguments of $f_i$ will always be $Pa(\mathrm{x}_i)$ throughout this paper. We highlight that $f_i$ does not depend on $i$.

**Identifiability.** We assume that the SCM follows an additive noise model (ANM) which is known to be identifiable from observational data (Hoyer et al., 2008; Peters et al., 2014). We also assume causal sufficiency, i.e. there are no hidden variables that are a common cause of at least two observed variables. In addition, corollary 33 from Peters et al. (2014) states that the true topological ordering of the DAG, as in our setting, is identifiable from a $p(\mathbf{x})$ generated by an ANM without requiring causal minimality assumptions.

**Finding Leaves with the Score.** Rolland et al. (2022) propose that the score of an ANM with distribution $p(\mathbf{x})$ can be used to find leaves[1]. Before presenting how to find the leaves, we derive, following Lemma 2 in Rolland et al. (2022), an analytical expression for the score which can be written as

$$
\begin{aligned}
\nabla_{\mathrm{x}_j} \log p(\mathbf{x}) &= \nabla_{\mathrm{x}_j} \log \prod_{i=1}^{d} p(\mathrm{x}_i \mid Pa(\mathrm{x}_i)) \\
&= \nabla_{\mathrm{x}_j} \sum_{i=1}^{d} \log p(\mathrm{x}_i \mid Pa(\mathrm{x}_i)) \\
&= \nabla_{\mathrm{x}_j} \sum_{i=1}^{d} \log p^\epsilon (\mathrm{x}_i - f_i) \qquad\qquad \triangleright \text{Using } \epsilon_i = \mathrm{x}_i - f_i \\
&= \frac{\partial \log p^\epsilon (\mathrm{x}_j - f_j)}{\partial \mathrm{x}_j} - \sum_{i \in Ch(\mathrm{x}_j)} \frac{\partial f_i}{\partial \mathrm{x}_j} \frac{\partial \log p^\epsilon (\mathrm{x}_i - f_i)}{\partial x}.
\end{aligned}
\tag{1}
$$

Where $Ch(\mathrm{x}_j)$ denotes the children of $\mathrm{x}_j$. We now proceed, based on Rolland et al. (2022), to derive a condition which can be used to find leaf nodes.

**Lemma 1.** *Given a nonlinear ANM with a noise distribution $p^\epsilon$ and a leaf node $j$; assume that $\frac{\partial^2 \log p^\epsilon}{\partial x^2} = a$, where $a$ is a constant, then*

$$
\mathrm{Var}_{\boldsymbol{X}} \left[ \boldsymbol{H}_{j,j}(\log p(\mathbf{x})) \right] = 0.
\tag{2}
$$

See proof in Appendix A.1.

**Remark.** *Lemma 1 enables finding leaf nodes based on the diagonal of the log-likelihood's Hessian.*

Rolland et al. (2022), using a similar conclusion, propose a topological ordering algorithm that iteratively finds and removes leaf nodes from the dataset. At each iteration Rolland et al. (2022) re-compute the Hessian with a kernel-based estimation method. In this paper, we develop a more efficient algorithm for learning the Hessian at high-dimensions and for a large number of samples. Note that Rolland et al. (2022) prove that Equation 2 can identify leaves in nonlinear ANMs with *Gaussian noise*. We derive a formulation which, instead, requires the second-order derivative of the noise distribution to be constant. Indeed, the condition $\frac{\partial^2 \log p^\epsilon}{\partial x^2} = a$ is true for $p^\epsilon$ following a Gaussian distribution which is consistent with Rolland et al. (2022), but could potentially be true for other distributions as well.

---

[1]We refer to nodes without children in a DAG $\mathcal{G}$ as leaves.

## 2.3 Diffusion Models Approximate the Score

The process of learning to denoise (Vincent, 2011) can approximate that of matching the score (Hyvärinen, 2005). A diffusion process gradually adds noise to a data distribution over time. Diffusion probabilistic models (DPMs) Sohl-Dickstein et al. (2015); Ho et al. (2020); Song et al. (2021) learn to reverse the diffusion process, starting with noise and recovering the data distribution. The diffusion process gradually adds Gaussian noise, with a time-dependent variance $\alpha_t$, to a sample $\mathbf{x}_0 \sim p_{\text{data}}(\mathbf{x})$ from the data distribution. Thus, the noisy variable $\mathbf{x}_t$, with $t \in [0, T]$, is learned to correspond to versions of $\mathbf{x}_0$ perturbed by Gaussian noise following $p(\mathbf{x}_t \mid \mathbf{x}_0) = \mathcal{N}(\mathbf{x}_t; \sqrt{\alpha_t}\mathbf{x}_0, (1 - \alpha_t)\boldsymbol{I})$, where $\alpha_t := \prod_{j=0}^{t}(1 - \beta_j)$, $\beta_j$ is the variance scheduled between $[\beta_{\min}, \beta_{\max}]$ and $\boldsymbol{I}$ is the identity matrix. DPMs (Ho et al., 2020) are learned with a weighted sum of denoising score matching objectives at different perturbation scales with

$$\theta^* = \arg\min_{\theta} \mathbb{E}_{\mathbf{x}_0, t, \epsilon} \left[ \lambda(t) \left\| \boldsymbol{\epsilon}_\theta(\mathbf{x}_t, t) - \epsilon \right\|_2^2 \right], \tag{3}$$

where $\mathbf{x}_t = \sqrt{\alpha_t}\mathbf{x}_0 + \sqrt{1 - \alpha_t}\epsilon$, with $\mathbf{x}_0 \sim p(\mathbf{x})$ being a sample from the data distribution, $t \sim \mathcal{U}(0, T)$ and $\epsilon \sim \mathcal{N}(0, \boldsymbol{I})$ is the noise. $\lambda(t)$ is a loss weighting term following Ho et al. (2020).

**Remark.** *Throughout this paper, we leverage the fact that the trained model $\boldsymbol{\epsilon}_\theta$ approximates the score $\nabla_{\mathbf{x}_j} \log p(\mathbf{x})$ of the data (Song & Ermon, 2019).*

## 3 The Deciduous Score

Discovering the complete topological ordering with the distribution's Hessian (Rolland et al., 2022) is done by finding the leaf node (Equation 2), appending the leaf node $\mathbf{x}_l$ to the ordering list $\pi$ and removing the data column corresponding to $\mathbf{x}_l$ from $\boldsymbol{X}$ before the next iteration $d - 1$ times. Rolland et al. (2022) estimate the score's Jacobian (Hessian) at each iteration.

Instead, we explore an alternative approach that does not require estimation of a new score after each leaf removal. In particular, we describe how to adjust the score of a distribution after each leaf removal, terming this a "deciduous score"[1]. We obtain an analytical expression for the deciduous score and derive a way of computing it, based on the original score before leaf removal. In this section, we only consider that $p(\mathbf{x})$ follows a distribution described by an ANM, we pose no additional assumptions over the noise distribution.

**Definition 1.** *Considering a DAG $\mathcal{G}$ which entails a distribution $p(\mathbf{x}) = \prod_{i=1}^{d} p(\mathbf{x}_i \mid Pa(\mathbf{x}_i))$. Let $p(\mathbf{x}_{-l}) = \frac{p(\mathbf{x})}{p(\mathbf{x}_l \mid Pa(\mathbf{x}_l))}$ be $p(\mathbf{x})$ without the random variable corresponding to the leaf node $\mathbf{x}_l$. The deciduous score $\nabla \log p(\mathbf{x}_{-l}) \in \mathbb{R}^{d-1}$ is the score of the distribution $p(\mathbf{x}_{-l})$.*

**Lemma 2.** *Given a ANM which entails a distribution $p(\mathbf{x})$, we can use Equation 1 to find an analytical expression for an additive residue $\Delta_l$ between the distribution's score $\nabla \log p(\mathbf{x})$ and its deciduous score $\nabla \log p(\mathbf{x}_{-l})$ such that*

$$\Delta_l = \nabla \log p(\mathbf{x}) - \nabla \log p(\mathbf{x}_{-l}). \tag{4}$$

*In particular, $\Delta_l$ is a vector $\{\delta_j \mid \forall j \in [1, \dots, d] \setminus l\}$ where the residue w.r.t. a node $\mathbf{x}_j$ can be denoted as*

$$\begin{aligned}
\delta_j &= \nabla_{\mathbf{x}_j} \log p(\mathbf{x}) - \nabla_{\mathbf{x}_j} \log p(\mathbf{x}_{-l}) \\
&= -\frac{\partial f_i}{\partial \mathbf{x}_j} \frac{\partial \log p^\epsilon(\mathbf{x}_i - f_i)}{\partial x}.
\end{aligned} \tag{5}$$

*If $\mathbf{x}_j \notin Pa(\mathbf{x}_l)$, $\delta_j = 0$.*

*Proof.* Observing Equation 1, the score $\nabla_{\mathbf{x}_j} \log p(\mathbf{x})$ only depends on the following random variables (i) $Pa(\mathbf{x}_j)$, (ii) $Ch(\mathbf{x}_j)$, and (iii) $Pa(Ch(\mathbf{x}_j))$. We consider $\mathbf{x}_l$ to be a leaf node, therefore $\nabla_{\mathbf{x}_j} \log p(\mathbf{x})$ only depends on $\mathbf{x}_l$ if $\mathbf{x}_j \in Pa(\mathbf{x}_l)$. If $\mathbf{x}_j \in Pa(\mathbf{x}_l)$, the only term depending on $\nabla_{\mathbf{x}_j} \log p(\mathbf{x})$ dependent on $\mathbf{x}_l$ is one of the terms inside the summation. $\qquad\square$

---

[1] An analogy to *deciduous* trees which seasonally shed leaves during autumn.

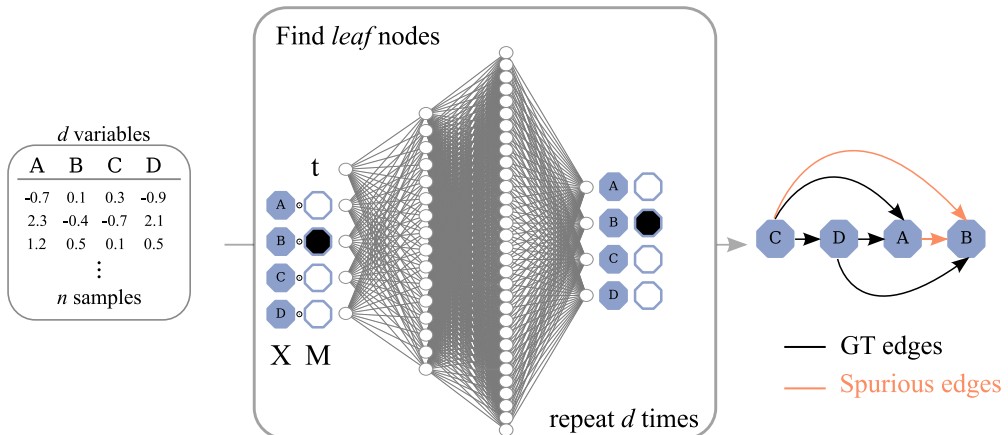

Figure 2: Topological ordering with diffusion models by iteratively finding leaf nodes. At each iteration, one leaf node is found using Equation 9. In the subsequent iteration, the previous leaves are removed, reducing the search space. After topological ordering (as illustrated on the right side), the presence of edges (causal mechanisms) between variables can be inferred such that parents of each variable are selected from the preceding variables in the ordered list. Spurious edges can be pruned with feature selection as a post-processing step (Bühlmann et al., 2014; Lachapelle et al., 2020; Rolland et al., 2022).

However, we wish to estimate the deciduous score $\nabla \log p(\mathbf{x}_{-l})$ without direct access to the function $f_l$, to its derivative, nor to the distribution $p^\epsilon$. Therefore, we now derive an expression for $\Delta_l$ using solely the score and the Hessian of $\log p(\mathbf{x})$.

**Theorem 1.** *Consider an ANM of distribution $p(\mathbf{x})$ with score $\nabla \log p(\mathbf{x})$ and the score's Jacobian $\boldsymbol{H}(\log p(\mathbf{x}))$. The additive residue $\Delta_l$ necessary for computing the deciduous score (as in Proposition 2) can be estimated with*

$$\Delta_l = \boldsymbol{H}_l(\log p(\mathbf{x})) \cdot \frac{\nabla_{\mathbf{x}_l} \log p(\mathbf{x})}{\boldsymbol{H}_{l,l}(\log p(\mathbf{x}))}. \tag{6}$$

See proof in Appendix A.2.

## 4 CAUSAL DISCOVERY WITH DIFFUSION MODELS

DPMs approximate the score of the data distribution (Song & Ermon, 2019). In this section, we explore how to use DPMs to perform leaf discovery and compute the deciduous score, based on Theorem 1, for iteratively finding and removing leaf nodes *without* re-training the score.

### 4.1 APPROXIMATING THE SCORE'S JACOBIAN VIA DIFFUSION TRAINING

The score's Jacobian can be approximated by learning the score $\boldsymbol{\epsilon}_\theta$ with denoising diffusion training of neural networks and back-propagating (Rumelhart et al., 1986)[1] from the output to the input variables. It can be written, for an input data point $\boldsymbol{x} \in \mathbb{R}^d$, as

$$\boldsymbol{H}_{i,j} \log p(\boldsymbol{x}) \approx \nabla_{i,j} \boldsymbol{\epsilon}_\theta(\boldsymbol{x}, t), \tag{7}$$

where $\nabla_{i,j} \boldsymbol{\epsilon}_\theta(\boldsymbol{x}, t)$ means the $ith$ output of $\boldsymbol{\epsilon}_\theta$ is backpropagated to the $jth$ input. The diagonal of the Hessian in Equation 7 can, then, be used for finding leaf nodes as in Equation 2.

In a two variable setting, it is sufficient for causal discovery to (i) train a diffusion model (Equation 3); (ii) approximate the score's Jacobian via backpropagation (Equation 7); (iii) compute variance of

---

[1]The Jacobian of a neural network can be efficiently computed with auto-differentiation libraries such as functorch (Horace He, 2021).

the diagonal across all data points; (iv) identify the variable with lowest variance as effect (Equation 2). We illustrate in Appendix C the Hessian of a two variable SCM computed with a diffusion model.

## 4.2 TOPOLOGICAL ORDERING

When a DAG contains more than two nodes, the process of finding leaf nodes (i.e. the topological order) needs to be done iteratively as illustrated in Figure 2. The naive (greedy) approach would be to remove the leaf node from the dataset, recompute the score, and compute the variance of the new distribution's Hessian to identify the next leaf node (Rolland et al., 2022). Since we employ diffusion models to estimate the score, this equates to re-training the model each time after a leaf is removed.

We hence propose a method to compute the deciduous score $\nabla \log p(\mathbf{x}_{-l})$ using Theorem 1 to remove leaves from the initial score *without* re-training the neural network. In particular, assuming that a leaf $\mathrm{x}_l$ is found, the residue $\Delta_l$ can be approximated[1] with

$$\Delta_l(\boldsymbol{x}, t) \approx \nabla_l \boldsymbol{\epsilon}_\theta(\boldsymbol{x}, t) \cdot \frac{\boldsymbol{\epsilon}_\theta(\boldsymbol{x}, t)_l}{\nabla_{l,l} \boldsymbol{\epsilon}_\theta(\boldsymbol{x}, t)} \tag{8}$$

where $\boldsymbol{\epsilon}_\theta(\boldsymbol{x}, t)_l$ is output corresponding to the leaf node. Note that the term $\nabla_l \boldsymbol{\epsilon}_\theta(\boldsymbol{x}, t)$ is a vector of size $d$ and the other term is a scalar. During topological ordering, we compute $\Delta_\pi$, which is the summation of $\Delta_l$ over all leaves already discovered and appended to $\pi$. Naturally, we only compute $\Delta_l$ w.r.t. nodes $\mathrm{x}_j \notin \pi$ because $\mathrm{x}_j \in \pi$ have already been ordered and are not taken into account anymore.

In practice, we observe that training $\boldsymbol{\epsilon}_\theta$ on $\boldsymbol{X}$ but using a subsample $\boldsymbol{B} \in \mathbb{R}^{k \times d}$ of size $k$ randomly sampled from $\boldsymbol{X}$ increases speed without compromising performance (see Section 4.3). In addition, analysing Equation 5, the absolute value of the residue $\delta_l$ decreases if the values of $\mathrm{x}_l$ are set to zero once the leaf node is discovered. Therefore, we apply a mask $\boldsymbol{M}_\pi \in \{0, 1\}^{k \times d}$ over leaves discovered in the previous iterations and compute only the Jacobian of the outputs corresponding to $\mathrm{x}_{-l}$. $\boldsymbol{M}_\pi$ is updated after each iteration based on the ordered nodes $\pi$. We then find a leaf node according to

$$\text{leaf} = \underset{\mathrm{x}_i \in \mathbf{x}}{\arg \min} \, \text{Var}_{\boldsymbol{B}} \left[ \nabla_{\mathbf{x}} \left( score(\boldsymbol{M}_\pi \odot \boldsymbol{B}, t) \right) \right], \tag{9}$$

where $\boldsymbol{\epsilon}_\theta$ is a DPM trained with Equation 3. See Appendix E.3 for the choice of $t$. This topological ordering procedure is formally described in Algorithm 1, $score(-\pi)$ means that we only consider the outputs for nodes $\mathrm{x}_j \notin \pi$

---

**Algorithm 1:** Topological Ordering with DiffAN

---

**Input:** $\boldsymbol{X} \in \mathbb{R}^{n \times d}$, trained diffusion model $\boldsymbol{\epsilon}_\theta$, ordering batch size $k$
$\pi = [], \Delta_\pi = \mathbf{0}^{k \times d}, \boldsymbol{M}_\pi = \mathbf{1}^{k \times d}, score = \boldsymbol{\epsilon}_\theta$
**while** $\|\pi\| \neq d$ **do**
    $\boldsymbol{B} \xleftarrow{k} \boldsymbol{X}$           // Randomly sample a batch of $k$ elements
    $\boldsymbol{B} \leftarrow \boldsymbol{B} \circ \boldsymbol{M}_\pi$           // Mask removed leaves
    $\Delta_\pi = \text{Get}\Delta_\pi(score, \boldsymbol{B})$           // Sum of Equation 8 over $\pi$
    $score = score(-\pi) + \Delta_\pi$           // Update score with residue
    $leaf = \text{GetLeaf}(score, \boldsymbol{B})$           // Equation 9
    $\pi = [leaf, \pi]$           // Append leaf to ordered list
    $\boldsymbol{M}_{:,leaf} = \mathbf{0}$           // Set discovered leaf to zero
**end**
**Output:** Topological order $\pi$

---

---

[1]The diffusion model itself is an approximation of the score, therefore its gradients are approximations of the score derivatives.

### 4.3 Computational Complexity and Practical Considerations

We now study the complexity of topological ordering with DiffAN w.r.t. the number of samples $n$ and number of variables $d$ in a dataset. In addition, we discuss what are the complexities of a greedy version as well as approximation which only utilises masking.

**Complexity on $n$.** Our method separates learning the score $\epsilon_\theta$ from computing the variance of the Hessian's diagonal across data points, in contrast to Rolland et al. (2022). We use all $n$ samples in $\boldsymbol{X}$ for learning the score function with diffusion training (Equation 3). It does *not* involve expensive constrained optimisation techniques[1] and we train the model for a fixed number of epochs (which is linear with $n$) or until reaching the early stopping criteria. We use a MLP that grows in width with $d$ but it does not significantly affect complexity. Therefore, we consider training to be $O(n)$. Moreover, Algorithm 1 is computed over a batch $\boldsymbol{B}$ with size $k < n$ instead of the entire dataset $\boldsymbol{X}$, as described in Equation 9. Note that the number of samples $k$ in $\boldsymbol{B}$ can be arbitrarily small and constant for different datasets. In Section 5.2, we verify that the accuracy of causal discovery initially improves as $k$ is increased but eventually tapers off.

**Complexity on $d$.** Once $\epsilon_\theta$ is trained, a topological ordering can be obtained by running $\nabla_{\boldsymbol{x}}\epsilon_\theta(\boldsymbol{x}, t)$ $d$ times. Moreover, computing the Jacobian of the score requires back-propagating the gradients $d - i$ times, where $i$ is the number of nodes already ordered in a given iteration. Finally, computing the deciduous score's residue (Equation 8) means computing gradient of the $i$ nodes. Resulting in a complexity of $O(d \cdot (d - i) \cdot i)$ with $i$ varying from 0 to $d$ which can be described by $O(d^3)$. The final topological ordering complexity is therefore $O(n + d^3)$.

**DiffAN Masking.** We verify empirically that the masking procedure described in Section 4.2 can significantly reduce the deciduous score's residue absolute value while maintaining causal discovery capabilities. In DiffAN Masking, we do not re-train the $\epsilon_\theta$ nor compute the deciduous score. This ordering algorithm is an approximation but has shown to work well in practice while showing remarkable scalability. DiffAN Masking has $O(n + d^2)$ ordering complexity.

## 5 Experiments

In our experiments, we train a NN with a DPM objective to perform topological ordering and follow this with a pruning post-processing step (Bühlmann et al., 2014). The performance is evaluated on synthetic and real data and compared to state-of-the-art causal discovery methods from observational data which are either ordering-based or gradient-based methods, **NN architecture.** We use a 4-layer multilayer perceptron (MLP) with LeakyReLU and layer normalisation. **Metrics.** We use the structural Hamming distance (SHD), Structural Intervention Distance (SID) (Peters & Bühlmann, 2015), *Order Divergence* (Rolland et al., 2022) and run time in seconds. See Appendix D.3 for details of each metric. **Baselines.** We use CAM (Bühlmann et al., 2014), GranDAG (Lachapelle et al., 2020) and SCORE (Rolland et al., 2022). We apply the pruning procedure of Bühlmann et al. (2014) to all methods. See detailed results in the Appendix D. Experiments with real data from Sachs (Sachs et al., 2005) and SynTReN (Van den Bulcke et al., 2006) datasets are in the Appendix E.1.

### 5.1 Synthetic Data

In this experiment, we consider causal relationships with $f_i$ being a function sampled from a Gaussian Process (GP) with radial basis function kernel of bandwidth one. We generate data from additive noise models which follow a Gaussian, Exponential or Laplace distributions with noise scales in the intervals $\{[0.4, 0.8], [0.8, 1.2], [1, 1]\}$, which are known to be identifiable (Peters et al., 2014). The causal graph is generated using the Erdös-Rényi (ER) (Erdős et al., 1960) and Scale Free (SF) (Bollobás et al., 2003) models. For a fixed number of nodes $d$, we vary the sparsity of the sampled graph by setting the average number of edges to be either $d$ or $5d$. We use the notation $[d][\text{graph type}][\text{sparsity}]$ for indicating experiments over different synthetic datasets. We show that DiffAN performs on par with baselines while being extremely fast, see Figure 3. We also explore the role of overfitting in Appendix E.2, the difference between DiffAN with masking only and the greedy

---

[1]Such as the Augmented Lagrangian method (Zheng et al., 2018; Lachapelle et al., 2020).

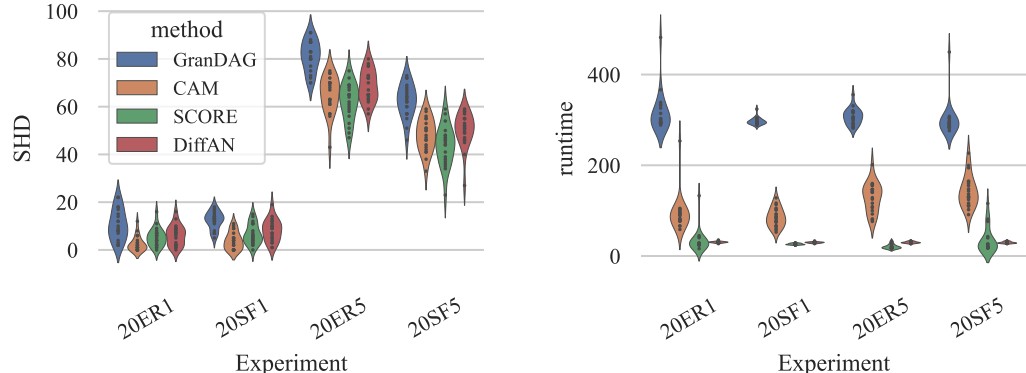

Figure 3: SHD (left) and run time in seconds (right) for experiments of synthetic data graphs for graphs with 20 nodes. The variation in the violinplots come from 3 different seeds over dataset generated from 3 different noise type and 3 different noise scales. Therefore, we run a total of 27 experiments for each method and synthetic datasets type

version in Appendix E.4, how we choose $t$ during ordering in Appendix E.3 and we give results stratified by experiment in Appendix E.5.

## 5.2 SCALING UP WITH DIFFAN MASKING

We now verify how DiffAN scales to bigger datasets, in terms of the number of samples $n$. Here, we use *DiffAN masking* because computing the residue with DiffAN would be too expensive for very big $d$. We evaluate only the topological ordering, ignoring the final pruning step. Therefore, the performance will be measured solely with the Order Divergence metric.

**Scaling to large datasets.** We evaluate how DiffAN compares to SCORE (Rolland et al., 2022), the previous state-of-the-art, in terms of run time (in seconds) and and the performance (order divergence) over datasets with $d = 500$ and different sample sizes $n \in 10^2, \ldots, 10^5$, the error bars are results across 6 dataset (different samples of ER and SF graphs). As illustrated in Figure 1, DiffAN is the more tractable option as the size of the dataset increases. SCORE relies on inverting a very large $n \times n$ matrix which is expensive in memory and computing for large $n$. Running SCORE for $d = 500$ and $n > 2000$ is intractable in a machine with 64Gb of RAM. Figure 4 (left) shows that, since DiffAN can learn from bigger datasets and therefore achieve better results as sample size increases.

**Ordering batch size.** An important aspect of our method, discussed in Section 4.3, that allows scalability in terms of $n$ is the separation between learning the score function $\epsilon_\theta$ and computing the Hessian variance across a batch of size $k$, with $k < n$. Therefore, we show empirically, as illustrated in Figure 4 (right), that decreasing $k$ does not strongly impact performance for datasets with $d \in 10, 20, 50$.

## 6 RELATED WORKS

**Ordering-based Causal Discovery.** The observation that a causal DAG can be partially represented with a topological ordering goes back to Verma & Pearl (1990). Searching the topological ordering space instead of searching over the space of DAGs has been done with greedy Markov Chain Monte Carlo (MCMC) (Friedman & Koller, 2003), greedy hill-climbing search (Teyssier & Koller, 2005), arc search (Park & Klabjan, 2017), restricted maximum likelihood estimators (Bühlmann et al., 2014), sparsest permutation (Raskutti & Uhler, 2018; Lam et al., 2022; Solus et al., 2021), and reinforcement learning (Wang et al., 2021). In linear additive models, Ghoshal & Honorio (2018); Chen et al. (2019) propose an approach, under some assumptions on the noise variances, to discover the causal graph by sequentially identifying leaves based on an estimation of the precision matrix.

**Hessian of the Log-likelihood.** Estimating $\boldsymbol{H}(\log p(\mathbf{x}))$ is the most expensive task of the ordering algorithm. Our baseline (Rolland et al., 2022) propose an extension of Li & Turner (2018) which

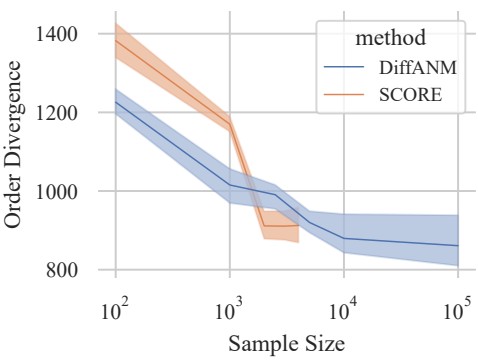 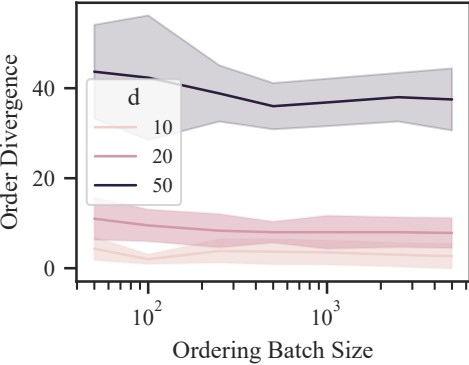

Figure 4: Accuracy of DiffAN as the dataset size is scaled up for datasets with 500 variables and increasing numbers of data samples $n$ (left) and as the batch size for computing the Hessian variance is changed (right). We show 95% confidence intervals over 6 datasets which have different graph structures sampled different graph types (ER/SF).

utilises the Stein's identity over a RBF kernel (Schölkopf & Smola, 2002). Rolland et al.'s method cannot obtain gradient estimates at positions out of the training samples. Therefore, evaluating the Hessian over a subsample of the training dataset is not possible. Other promising kernel-based approaches rely on spectral decomposition (Shi et al., 2018) solve this issue and can be promising future directions. Most importantly, computing the kernel matrix is expensive for memory and computation on $n$. There are, however, methods (Achlioptas et al., 2001; Halko et al., 2011; Si et al., 2017) that help scaling kernel techniques, which were not considered in the present work. Other approaches are also possible with deep likelihood methods such as normalizing flows (Durkan et al., 2019; Dinh et al., 2017) and further compute the Hessian via backpropagation. This would require two backpropagation passes giving $O(d^2)$ complexity and be less scalable than denoising diffusion. Indeed, preliminary experiments proved impractical in our high-dimensional settings.

We use DPMs because they can efficiently approximate the Hessian with a single backpropagation pass and while allowing Hessian evaluation on a subsample of the training dataset. It has been shown (Song & Ermon, 2019) that denoising diffusion can better capture the score than simple denoising (Vincent, 2011) because noise at multiple scales explore regions of low data density.

## 7 CONCLUSION

We have presented a scalable method using DPMs for causal discovery. Since DPMs approximate the score of the data distribution, they can be used to efficiently compute the log-likelihood's Hessian by backpropagating each element of the output with respect to each element of the input. The *deciduous score* allows adjusting the score to remove the contribution of the leaf most recently removed, avoiding re-training the NN. Our empirical results show that neural networks can be efficiently used for topological ordering in high-dimensional graphs (up to 500 nodes) and with large datasets (up to $10^5$ samples).

Our *deciduous score* can be used with other Hessian estimation techniques as long as obtaining the score and its full Jacobian is possible from a trained model, *e.g.* sliced score matching (Song et al., 2020) and approximate backpropagation (Kingma & Cun, 2010). Updating the score is more practical than re-training in most settings with neural networks. Therefore, our theoretical result enables the community to efficiently apply new score estimation methods to topological ordering. Moreover, DPMs have been previously used generative diffusion models in the context of causal estimation (Sanchez & Tsaftaris, 2022). In this work, we have not explored the generative aspect such as Geffner et al. (2022) does with normalising flows. Finally, another promising direction involves constraining the NN architecture as in Lachapelle et al. (2020) with constrained optimisation losses (Zheng et al., 2018).

## 8 ACKNOWLEDGEMENT

This work was supported by the University of Edinburgh, the Royal Academy of Engineering and Canon Medical Research Europe via P. Sanchez's PhD studentship. S.A. Tsaftaris acknowledges the support of Canon Medical and the Royal Academy of Engineering and the Research Chairs and Senior Research Fellowships scheme (grant RCSRF1819\825).

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

## A  PROOFS

We re-write Equation 1 here for improved readability:

$$\nabla_{\mathbf{x}_j} \log p(\mathbf{x}) = \frac{\partial \log p^\epsilon (\mathbf{x}_j - f_j)}{\partial \mathbf{x}_j} - \sum_{i \in Ch(\mathbf{x}_j)} \frac{\partial f_i}{\partial \mathbf{x}_j} \frac{\partial \log p^\epsilon (\mathbf{x}_i - f_i)}{\partial x}. \tag{10}$$

### A.1  PROOF LEMMA 1

*Proof.* We start by showing the "⇐" direction by deriving Equation 1 w.r.t. $\mathbf{x}_j$. If $\mathbf{x}_j$ is a leaf, only the first term of the equation is present, then taking its derivative results in

$$\begin{aligned} \boldsymbol{H}_{l,l}(\log p(\mathbf{x})) &= \frac{\partial^2 \log p^\epsilon (\mathbf{x}_l - f_l)}{\partial x^2} \cdot \frac{df_l}{d\mathbf{x}_l} \\ &= \frac{\partial^2 \log p^\epsilon (\mathbf{x}_l - f_l)}{\partial x^2}. \end{aligned} \tag{11}$$

Therefore, only if $j$ is a leaf and $\frac{d \log p^\epsilon}{dx^2} = a$, $\mathrm{Var}_{\boldsymbol{X}} [\boldsymbol{H}_{l,l}(\log p(\mathbf{x}))] = 0$. The remaining of the proof follows Rolland et al. (2022) (which was done for a Gaussian noise only), we prove by contradiction that ⇒ is also true. In particular, if we consider that $\mathbf{x}_j$ is **not** a leaf and $\boldsymbol{H}_{j,j} \log p(\mathbf{x}) = c$, with $c$ being a constant, we can write

$$\nabla_{\mathbf{x}_j} \log p(\mathbf{x}) = c\mathbf{x}_j + g(\mathbf{x}_{-j}). \tag{12}$$

Replacing Equation 12 in to Equation 1, we have

$$c\mathbf{x}_j + g(\mathbf{x}_{-j}) = \frac{\partial \log p^\epsilon (\mathbf{x}_j - f_j)}{\partial x} - \sum_{\mathbf{x}_i \in Ch(\mathbf{x}_j)} \frac{\partial f_i}{\partial \mathbf{x}_j} \frac{\partial \log p^\epsilon (\mathbf{x}_i - f_i)}{\partial x}. \tag{13}$$

Let $\mathbf{x}_c \in Ch(\mathbf{x}_j)$ such that $\mathbf{x}_c \notin Pa(Ch(\mathbf{x}_j))$. $\mathbf{x}_c$ always exist since $\mathbf{x}_j$ is not a leaf, and it suffices to pick a child of $\mathbf{x}_c$ appearing at last position in some topological order. If we isolate the terms depending on $\mathbf{x}_c$ on the RHS of Equation 13, we have

$$c\mathbf{x}_j + \frac{\partial \log p^\epsilon (\mathbf{x}_j - f_j)}{\partial x} - \sum_{\mathbf{x}_i \in Ch(\mathbf{x}_j), \mathbf{x}_i \neq \mathbf{x}_c} \frac{\partial f_i}{\partial \mathbf{x}_j} \frac{\partial \log p^\epsilon (\mathbf{x}_i - f_i)}{\partial x} = \frac{\partial f_c}{\partial \mathbf{x}_j} \frac{\partial \log p^\epsilon (\mathbf{x}_c - f_c)}{\partial x} - g(\mathbf{x}_{-j}). \tag{14}$$

Deriving both sides w.r.t. $\mathbf{x}_c$, since the LHS of Equation 14 does not depend on $\mathbf{x}_c$, we can write

$$\begin{aligned} &\frac{\partial}{\partial \mathbf{x}_c} \left( \frac{\partial f_c}{\partial \mathbf{x}_j} \frac{\partial \log p^\epsilon (\mathbf{x}_c - f_c)}{\partial x} - g(\mathbf{x}_{-j}) \right) = 0 \\ &\Rightarrow \frac{\partial f_c}{\partial \mathbf{x}_j} \frac{\partial \log p^\epsilon (\mathbf{x}_c - f_c)}{\partial x^2}^a = \frac{\partial g(\mathbf{x}_{-j})}{\partial \mathbf{x}_c} \end{aligned} \tag{15}$$

Since $g$ does not depend on $\mathbf{x}_j$, $\frac{\partial f_c}{\partial \mathbf{x}_j}$ does not depend on $\mathbf{x}_j$ neither, implying that $f_c$ is linear in $\mathbf{x}_j$, contradicting the non-linearity assumption. □

### A.2  PROOF THEOREM 1

*Proof.* Using Equation 1, we will derive expressions for each of the elements in Equation 6 and show that it is equivalent to $\delta_l$ in Equation 5. First, note that the score of a leaf node $\mathbf{x}_l$ can be denoted as:

$$\begin{aligned} \nabla_{\mathbf{x}_l} \log p(\mathbf{x}) &= \frac{\partial \log p^\epsilon (\mathbf{x}_j - f_j)}{\partial \mathbf{x}_j} \\ &= \frac{\partial \log p^\epsilon (\mathbf{x}_l - f_l)}{\partial x} \cdot \frac{d (\mathbf{x}_l - f_l)}{d\mathbf{x}_l} \\ &= \frac{\partial \log p^\epsilon (\mathbf{x}_l - f_l)}{\partial x} \end{aligned} \tag{16}$$

Second, replacing Equation 16 into each element of $\boldsymbol{H}_l(\log p(\mathbf{x})) \in R^d$, we can write

$$
\begin{aligned}
\boldsymbol{H}_{l,j}(\log p(\mathbf{x})) &= \frac{\partial}{\partial \mathbf{x}_j} \left[ \nabla_{\mathbf{x}_l} \log p(\mathbf{x}) \right] \\
&= \frac{\partial^2 \log p^\epsilon (\mathbf{x}_l - f_l)}{\partial x^2} \cdot \frac{d(\mathbf{x}_l - f_l)}{d\mathbf{x}_j} \\
&= \frac{\partial^2 \log p^\epsilon (\mathbf{x}_l - f_l)}{\partial x^2} \cdot \frac{df_l}{d\mathbf{x}_j}.
\end{aligned}
\tag{17}
$$

If $j = l$ in Equation 17, we have

$$
\begin{aligned}
\boldsymbol{H}_{l,l}(\log p(\mathbf{x})) &= \frac{\partial^2 \log p^\epsilon (\mathbf{x}_l - f_l)}{\partial x^2} \cdot \frac{df_l}{d\mathbf{x}_l} \\
&= \frac{\partial^2 \log p^\epsilon (\mathbf{x}_l - f_l)}{\partial x^2}.
\end{aligned}
\tag{18}
$$

Finally, replacing Equations 16, 17 and 11 into the Equation 5 for a single node $\mathbf{x}_j$, if $j \neq l$, we have

$$
\begin{aligned}
\delta_j &= \frac{\boldsymbol{H}_{l,j}(\log p(\mathbf{x})) \cdot \nabla_{\mathbf{x}_l} \log p(\mathbf{x})}{\boldsymbol{H}_{l,l}(\log p(\mathbf{x}))} \\
&= \frac{\frac{\partial \log p^\epsilon (\mathbf{x}_l - f_l)}{\partial x^2} \cdot \frac{df_l}{d\mathbf{x}_j} \cdot \frac{\partial \log p^\epsilon (\mathbf{x}_l - f_l)}{\partial x}}{\frac{\partial \log p^\epsilon (\mathbf{x}_l - f_l)}{\partial x^2}} \\
&= \frac{df_l}{d\mathbf{x}_j} \cdot \frac{\partial \log p^\epsilon (\mathbf{x}_l - f_l)}{\partial x}.
\end{aligned}
\tag{19}
$$

The last line in Equation 19 is the same as in Equation 5 from Lemma 2, proving that $\delta_j$ can be written using the first and second order derivative of the log-likelihood. $\qquad\square$

## B  SCORE OF NONLINEAR ANM WITH GAUSSIAN NOISE

A SCM entails a distribution

$$
p(\mathbf{x}) = \prod_{i=1}^{d} p(\mathbf{x}_i \mid Pa(\mathbf{x}_i)),
\tag{20}
$$

over the variables $\mathbf{x}$ (Peters et al., 2017) By assuming that the noise variables $\epsilon_i \sim \mathcal{N}(0, \sigma_i^2)$ and inserting the ANM function, Equation 20 can be written as

$$
\log p(\mathbf{x}) = -\frac{1}{2} \sum_{i=1}^{d} \left( \frac{\mathbf{x}_i - f_i(Pa(\mathbf{x}_i))}{\sigma_i} \right)^2 - \frac{1}{2} \sum_{i=1}^{d} \log(2\pi\sigma_i^2).
\tag{21}
$$

The score of $p(\mathbf{x})$ can hence be written as

$$
\nabla_{\mathbf{x}_j} \log p(\mathbf{x}) = -\frac{\mathbf{x}_j - f_j(Pa(\mathbf{x}_j))}{\sigma_j^2} + \sum_{i \in \text{children}(j)} \frac{\partial f_i}{\partial \mathbf{x}_j}(Pa(\mathbf{x}_i)) \frac{\mathbf{x}_i - f_i(Pa(\mathbf{x}_i))}{\sigma_i^2}.
\tag{22}
$$

## C  VISUALISATION OF THE SCORE'S JACOBIAN FOR TWO VARIABLES

Considering a two variables problem where the causal mechanisms are $B = f_\omega(A) + \epsilon_B$ and $A = \epsilon_A$ with $\epsilon_A, \epsilon_B \sim \mathcal{N}(0, 1)$ and $f_\omega$ being a two-layer MLP with randomly initialised weights. Note that, in Figure 5, the variance of $\frac{\partial^2 \log p(\boldsymbol{A}, \boldsymbol{B})}{\partial B^2}$, while not 0 as predicted by Equation 2, is smaller than $\frac{\partial^2 \log p(\boldsymbol{A}, \boldsymbol{B})}{\partial \boldsymbol{A}^2}$ allowing discovery of the true causal direction.

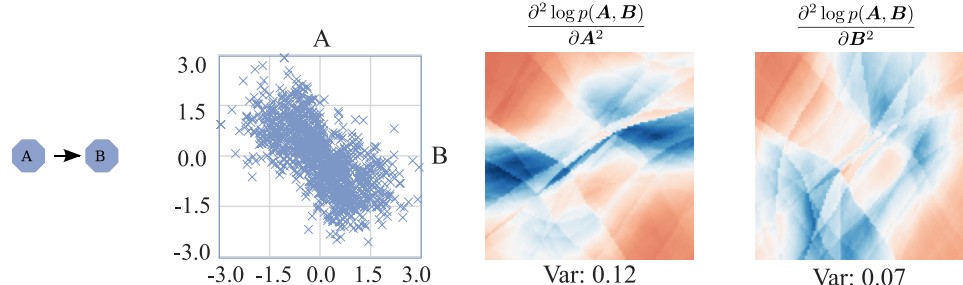

Figure 5: Visualisation of the diagonal of the score's Jacobian estimated with a diffusion model as in Equation 7 for a two-variable SCM where $A \rightarrow B$.

Table 1: MLP architecture. The hyperparameters of each Linear layer depend on $d$ such that big $= max(1024, 5 * d)$ and small $= max(128, 3 * d)$.

| Layer | Hyperparameters |
|---|---|
| Linear | Ch: $(d + 1, \text{small})$ |
| LeakyReLU | |
| LayerNorm | |
| Dropout | Prob: 0.2 |
| Linear | Ch: (small, big) |
| LeakyReLU | |
| LayerNorm | |
| Linear | Ch: (big, big) |
| LeakyReLU | |
| Linear | Ch: (big, big) |
| LeakyReLU | |
| Linear | Ch: (big, $d$) |

## D  EXPERIMENTS DETAILS

### D.1  HYPERPARAMETERS OF DPM TRAINING

We now describe the hyperparameters for the diffusion training. We use number of time steps $T = 100$, $\beta_t$ is a linearly scheduled between $\beta_{\min} = 0.0001$ and $\beta_{\max} = 0.02$. The model is trained according to Equation 3 which follows Ho et al. (2020). During sampling, $t$ is sampled from a Uniform distribution.

### D.2  NEURAL ARCHITECTURE

The neural network follows a simple MLP with 5 Linear layers, LeakyReLU activation function, Layer Normalization and Dropout in the first layer. The full architecture is detailed in Table 1.

### D.3  METRICS

For each method, we compute the

**SHD.** Structural Hamming distance between the output and the true causal graph, which counts the number of missing, falsely detected, or reversed edges.

**SID.** Structural Intervention Distance is based on a graphical criterion only and quantifies the closeness between two DAGs in terms of their corresponding causal inference statements(Peters & Bühlmann, 2015).

**Order Divergence.** Rolland et al. (2022) propose this quantity for measuring how well the topological order is estimated. For an ordering $\pi$, and a target adjacency matrix $A$, we define the topological

order divergence $D_{top}(\pi, A)$ as

$$D_{top}(\pi, \boldsymbol{A}) = \sum_{i=1}^{d} \sum_{j:\pi_i > \pi_j} \boldsymbol{A}_{ij}. \tag{23}$$

If $\pi$ is a correct topological order for $\boldsymbol{A}$, then $D_{top}(\pi, \boldsymbol{A}) = 0$. Otherwise, $D_{top}(\pi, \boldsymbol{A})$ counts the number of edges that cannot be recovered due to the choice of topological order. Therefore, it provides a lower bound on the SHD of the final algorithm (irrespective of the pruning method).

# E    OTHER RESULTS

## E.1    REAL DATA

We consider two real datasets: (i) Sachs: A protein signaling network based on expression levels of proteins and phospholipids (Sachs et al., 2005). We consider only the observational data ($n = 853$ samples) since our method targets discovery of causal mechanisms when only observational data is available. The ground truth causal graph given by Sachs et al. (2005) has 11 nodes and 17 edges. (ii) SynTReN: We also evaluate the models on a pseudo-real dataset sampled from SynTReN generator (Van den Bulcke et al., 2006). Results, in Table 2, show that our method is competitive against other state-of-the-art causal discovery baselines on real datasets.

Table 2: SHD and SID results over real datasets.

|  | Sachs | | SynTReN | |
| --- | --- | --- | --- | --- |
|  | SHD | SID | SHD | SID |
| CAM | 12 | 55 | 40.5 | 152.3 |
| GraN-DAG | 13 | 47 | 34.0 | 161.7 |
| SCORE | 12 | 45 | 36.2 | 193.4 |
| DiffAN (ours) | 13 | 56 | 39.7 | 173.5 |

## E.2    OVERFITTING

The data used for topological ordering (inference) is a subset of the training data. Therefore, it is not obvious if overfitting would be an issue with our algorithm. Therefore, we run an experiment where we fix the number of epochs to 2000 considered high for a set of runs and use early stopping for another set in order to verify if overfitting is an issue. On average across all 20 nodes datasets, the early stopping strategy output an ordering diverge of $9.5$ whilst overfitting is at $11.1$ showing that the method does not benefit from overfitting.

## E.3    OPTIMAL $t$ FOR SCORE ESTIMATION

As noted by Vincent (2011), the best approximation of the score by a learned denoising function is when the training signal-to-noise (SNR) ratio is low. In diffusion model training, $t = 0$ corresponds to the coefficient with lowest SNR. However, we found empirically that the best score estimate varies somehow randomly across different values of $t$. Therefore, we run the the leaf finding function (Equation 9) $N$ times for values of $t$ evenly spaced in the $[0, T]$ interval and choose the best leaf based on majority vote. We show in Figure 6 that majority voting is a better approach than choosing a constant value for $t$.

## E.4    ABLATIONS OF DIFFAN MASKING AND GREEDY

We now verify how DiffAN masking and DiffAN greedy compare against the original version detailed in the main text which computes the deciduous score. Here, we use the same datasets decribed in Section 5.1 which comprises 4 (20ER1, 20ER5, 20SF1, 20SF5) synthetic dataset types with 27 variations over seeds, noise type and noise scale.

**DiffAN Greedy.** A greedy version of the algorithm re-trains the $\epsilon_\theta$ after each leaf removal iteration. In this case, the deciduous score is not computed, decreasing the complexity w.r.t. $d$ but increasing w.r.t. $n$. DiffAN greedy has $O(nd^2)$ ordering complexity.

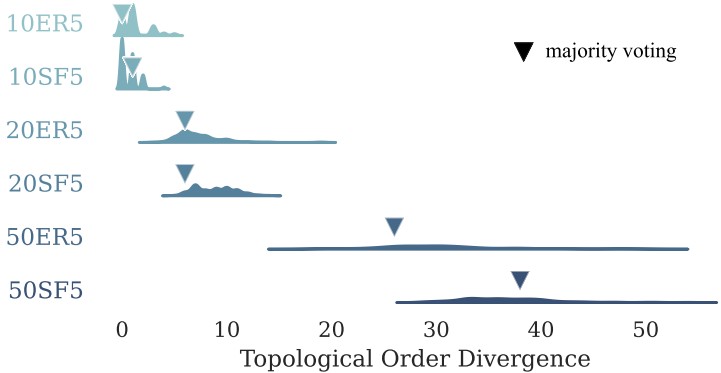

Figure 6: The distribution of order divergence measured for different values of $t$ is highly variable. Therefore, we show that we obtain a better approximation with majority voting.

We observe, in Figure 7, that the greedy version performs the best but it is the slowest, as seen in Figure 8. DiffAN masking

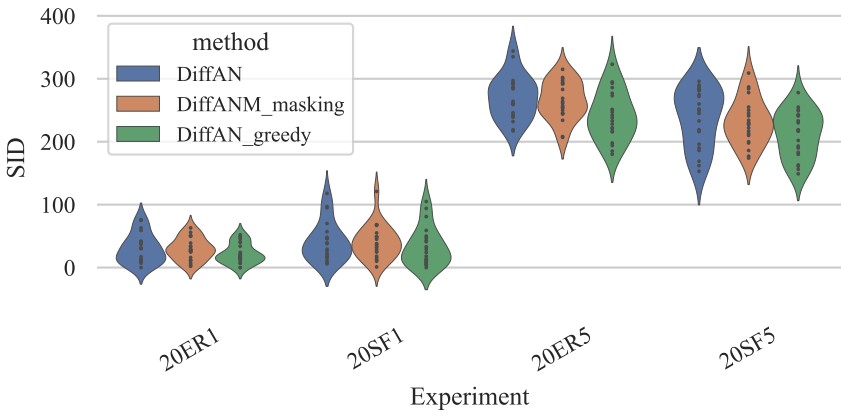

Figure 7: SID metric for different versions of DiffAN.

### E.5 DETAILED RESULTS

We present the numerical results for the violinplots in Section 5.1 in Tables 3 and 4. The results are presented in mean$_{std}$ with statistics acquired over experiments with 3 seeds.

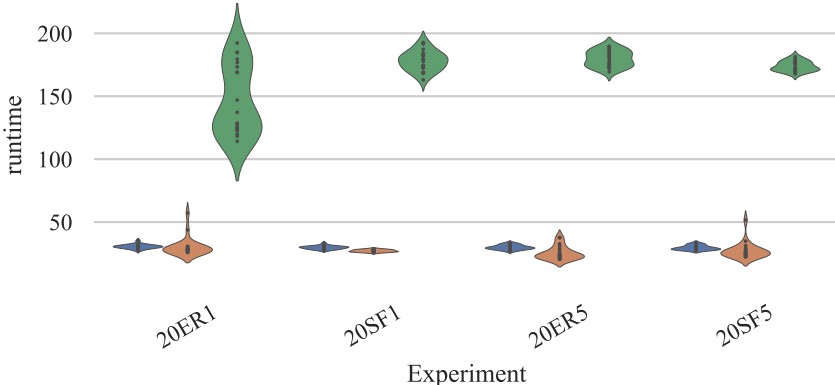

Figure 8: Runtime in seconds for different versions of DiffAN.

Table 3: Erdös-Rényi (ER) graphs.

| Exp Name | noisetype | method | shd | sid | runtime (seconds) |
|---|---|---|---|---|---|
| 20ER1 | exp | CAM | $6.50_{3.21}$ | $35.50_{27.54}$ | $112.17_{69.57}$ |
| | | DiffAN | $7.67_{4.68}$ | $38.83_{24.94}$ | $30.07_{1.60}$ |
| | | DiffAN Greedy | $6.67_{4.68}$ | $34.17_{18.37}$ | $121.66_{4.19}$ |
| | | GranDAG | $12.83_{5.60}$ | $65.17_{21.99}$ | $341.69_{74.82}$ |
| | | SCORE | $4.17_{2.86}$ | $17.50_{11.26}$ | $50.49_{41.87}$ |
| | gauss | CAM | $1.33_{1.03}$ | $6.50_{6.69}$ | $83.27_{11.75}$ |
| | | DiffAN | $8.17_{4.02}$ | $41.67_{24.21}$ | $30.98_{2.28}$ |
| | | DiffAN Greedy | $3.33_{2.16}$ | $18.33_{11.40}$ | $129.10_{9.54}$ |
| | | GranDAG | $10.50_{8.62}$ | $44.50_{37.90}$ | $304.07_{17.39}$ |
| | | SCORE | $8.67_{4.13}$ | $41.67_{21.54}$ | $24.77_{0.20}$ |
| | laplace | CAM | $0.83_{0.98}$ | $4.00_{6.96}$ | $90.85_{15.49}$ |
| | | DiffAN | $2.33_{2.07}$ | $8.67_{4.68}$ | $31.01_{1.18}$ |
| | | DiffAN Greedy | $3.00_{1.67}$ | $14.33_{4.76}$ | $175.30_{19.71}$ |
| | | GranDAG | $10.33_{5.82}$ | $41.17_{24.98}$ | $302.00_{16.23}$ |
| | | SCORE | $4.17_{3.43}$ | $24.17_{19.57}$ | $26.62_{1.88}$ |
| 20ER5 | exp | CAM | $60.67_{9.77}$ | $240.00_{33.02}$ | $105.65_{27.49}$ |
| | | DiffAN | $67.50_{4.32}$ | $278.83_{22.83}$ | $30.33_{2.12}$ |
| | | DiffAN Greedy | $63.00_{6.07}$ | $266.33_{50.25}$ | $186.21_{3.28}$ |
| | | GranDAG | $80.17_{7.99}$ | $304.17_{26.90}$ | $310.24_{13.71}$ |
| | | SCORE | $55.83_{7.55}$ | $190.67_{32.91}$ | $18.41_{3.73}$ |
| | gauss | CAM | $64.67_{5.85}$ | $214.67_{11.11}$ | $130.82_{26.10}$ |
| | | DiffAN | $68.50_{8.12}$ | $289.00_{42.31}$ | $29.58_{1.99}$ |
| | | DiffAN Greedy | $63.17_{6.18}$ | $228.83_{32.86}$ | $178.17_{7.17}$ |
| | | GranDAG | $81.00_{6.72}$ | $273.00_{46.72}$ | $319.17_{18.81}$ |
| | | SCORE | $63.50_{6.69}$ | $233.83_{39.65}$ | $21.29_{6.63}$ |
| | laplace | CAM | $68.00_{7.04}$ | $228.50_{27.86}$ | $157.70_{22.55}$ |
| | | DiffAN | $68.83_{7.86}$ | $248.67_{30.36}$ | $29.52_{1.73}$ |
| | | DiffAN Greedy | $67.33_{8.55}$ | $238.00_{35.19}$ | $176.71_{4.80}$ |
| | | GranDAG | $82.67_{6.47}$ | $271.33_{35.19}$ | $305.29_{13.93}$ |
| | | SCORE | $66.33_{7.09}$ | $218.67_{25.81}$ | $23.01_{4.33}$ |

Table 4: Scale Free (SF) graphs.

| Exp Name | noisetype | method | shd | sid | runtime (seconds) |
|---|---|---|---|---|---|
| 20SF1 | exp | CAM | $7.17_{2.86}$ | $32.67_{19.54}$ | $85.08_{27.71}$ |
| | | DiffAN | $8.67_{2.80}$ | $38.67_{20.99}$ | $29.23_{1.36}$ |
| | | DiffAN Greedy | $9.17_{2.32}$ | $35.33_{17.26}$ | $183.80_{4.81}$ |
| | | GranDAG | $15.50_{2.59}$ | $68.67_{24.19}$ | $298.88_{4.96}$ |
| | | SCORE | $5.83_{5.04}$ | $23.50_{23.65}$ | $26.13_{1.39}$ |
| | gauss | CAM | $1.83_{2.23}$ | $9.17_{10.21}$ | $84.46_{19.07}$ |
| | | DiffAN | $9.83_{6.24}$ | $56.67_{41.59}$ | $30.19_{0.92}$ |
| | | DiffAN Greedy | $7.33_{5.32}$ | $46.83_{38.42}$ | $175.63_{8.82}$ |
| | | GranDAG | $12.17_{2.71}$ | $46.67_{16.22}$ | $299.25_{13.14}$ |
| | | SCORE | $8.17_{4.45}$ | $41.67_{28.75}$ | $26.29_{1.83}$ |
| | laplace | CAM | $2.83_{3.92}$ | $7.83_{10.93}$ | $85.15_{21.84}$ |
| | | DiffAN | $6.00_{3.74}$ | $24.17_{16.18}$ | $29.68_{2.00}$ |
| | | DiffAN Greedy | $4.83_{3.66}$ | $17.17_{17.47}$ | $177.05_{8.82}$ |
| | | GranDAG | $9.50_{3.27}$ | $35.67_{20.84}$ | $295.12_{2.88}$ |
| | | SCORE | $5.67_{3.50}$ | $22.50_{15.27}$ | $26.39_{1.79}$ |
| 20SF5 | exp | CAM | $47.83_{5.78}$ | $228.83_{53.89}$ | $113.82_{13.32}$ |
| | | DiffAN | $46.83_{11.48}$ | $243.50_{34.68}$ | $30.03_{2.17}$ |
| | | DiffAN Greedy | $43.50_{6.89}$ | $236.83_{23.17}$ | $173.13_{3.52}$ |
| | | GranDAG | $60.67_{8.80}$ | $275.00_{22.56}$ | $284.34_{5.33}$ |
| | | SCORE | $38.50_{9.14}$ | $180.33_{57.44}$ | $19.79_{2.92}$ |
| | gauss | CAM | $46.83_{8.11}$ | $199.17_{53.13}$ | $130.41_{20.55}$ |
| | | DiffAN | $50.83_{4.62}$ | $259.50_{47.54}$ | $28.58_{1.16}$ |
| | | DiffAN Greedy | $45.17_{6.15}$ | $224.17_{41.80}$ | $174.90_{3.69}$ |
| | | GranDAG | $61.67_{4.03}$ | $241.67_{43.11}$ | $292.77_{7.29}$ |
| | | SCORE | $44.50_{5.24}$ | $217.83_{50.05}$ | $19.77_{2.51}$ |
| | laplace | CAM | $49.50_{9.01}$ | $191.33_{27.43}$ | $160.53_{23.16}$ |
| | | DiffAN | $52.00_{5.66}$ | $230.67_{50.66}$ | $29.05_{1.60}$ |
| | | DiffAN Greedy | $47.00_{9.94}$ | $191.33_{26.16}$ | $173.42_{3.52}$ |
| | | GranDAG | $65.00_{7.13}$ | $262.50_{44.23}$ | $295.40_{7.72}$ |
| | | SCORE | $46.67_{10.03}$ | $193.83_{34.14}$ | $43.84_{27.14}$ |

