# OpenReview forum: "Diffusion Models for Causal Discovery via Topological Ordering"
_ICLR.cc/2023/Conference — ICLR 2023 poster_

### Official Review · Reviewer_vVJW · 2022-10-17

**Confidence:** 3
**Correctness:** 4
**Technical Novelty And Significance:** 3
**Empirical Novelty And Significance:** Not applicable
**Recommendation:** 6

**Clarity, Quality, Novelty And Reproducibility:**

While the paper makes two solid technical contributions, they are somewhat obscured by the paper’s overemphasis on diffusion models (currently a hot topic in generative modeling), which have more to do with the implementation that the novel contributions. Ultimately, the paper (i) extends the results from Rolland et al. (2022) to not require Gaussianity assumptions and (ii) introduces a more efficient way of computing the elements along the diagonal once given access to the score function using backpropagation. While this is described as “diffusion models for causal discovery”, diffusion models are only leveraged for their estimates of the score function (as opposed to their generative modeling capabilities) - any nonparametric density estimation approach (with backprop) could be used in place of the diffusion model. Ultimately, this is an advantage of the approach, but the treatment of diffusion models as a necessity related to the contribution (rather than a means for implementation) impacts the overall clarity.

**Strength And Weaknesses:**

Strengths:
- New theoretical insights are provided, which generalize the results from Rolland et al. (2022)
- A more efficient algorithm is provided
- Empirical results demonstrate the gain in efficiency

Weaknesses:
- Too much emphasis on the diffusion models aspect makes the contribution confusing (see below)
- More explicit connections between the technical results and the correctness of the algorithm would improve clarity

**Summary Of The Paper:**

The paper proposes a more efficient approach for finding a topological ordering based on identifying low variance entries along the diagonal of the Hessian of the log density. Unlike the previous approach, this method does not require iteratively reestimating the entire Hessian to identify each leaf node. The approach is implemented using diffusion models, since they approximate the score function. The resulting method performs comparably to the previous state of the art but is more efficient in high dimensions.

**Summary Of The Review:**

The paper proposes an improvement to causal discovery based on estimating the variance of the Hessian log density diagonal that generalizes the causal discovery method and improves efficiency. The paper makes two core technical contributions and empirical results demonstrate the gain in efficiency. The clarity of the paper could be improved to emphasize the specific contributions made and the role (and necessity) of diffusion models in the proposed approach.

---

> ### Author Response · Authors · 2022-11-11
> **Response to vVJW**
>
> We thank the reviewer for the insightful comments and we appreciate that the reviewer agrees on the importance of our core technical contributions.
>
> We appreciate your comments pointing out that the contributions should be clearer. We have updated the paper accordingly, showing that using NNs in this task allows for better scalability. Indeed, the deciduous score enables the ML community to, in principle, efficiently apply any score estimation methods  to topological ordering.
>
> ### Q: “More explicit connections between the technical results and the correctness of the algorithm would improve clarity”
>
> Our algorithm enjoy better scalability because the “deciduous score” update rule avoid re-training.  Re-training the NNs, especially for large datasets, would be too onerous.  Our expanded scalability experiments (see page Figure 4, left) show that after increasing the depth of the NNs, DiffAN can outperform the baseline. This highlights explicitly that the algorithm does improve performance without need for re-training.
>
> We highlight that our focus is to devise a method that scales well. We now emphasise this in the paper profoundly.
>
> ### Q: “any nonparametric density estimation approach (with backprop) could be used in place of the diffusion model ”
>
> We agree that other density estimation methods could be used instead of DPMs. Indeed, we performed preliminary experiments with normalising flows (NFs). We concluded that they were less suited than diffusion models for this particular task.
> We refer the reviewer to the global response for an in-depth discussion of why DPMs are crucial to our method.
>
> ### Q: “treatment of diffusion models as a necessity”
> As discussed in the general comment, DPMs are not a necessity but we believe that they are the best tool for this setting. We refer the reviewer to the “Why diffusion models” section of the global response.
>
> ### Q: “as opposed to their generative modeling capabilities”
> Exploring their generative capabilities is an exciting future work direction, as we described in the updated version of the conclusion. Recently, some causal discovery methods [1] leverage generative models (NFs) along with constrained optimisation losses. We envision that DPMs could potentially be in similar settings as well. We add this discussion in the conclusion to encourage further exploration by the community.
>
> ### References
>
> [1] Tomas Geffner et al. Deep End-to-end Causal Inference. 2022.
>
> [2] Xun Zhenget al DAGs with NO TEARS: Continuous Optimization for Structure Learning. Neurips 2018.

---

### Official Review · Reviewer_YaE5 · 2022-10-24

**Confidence:** 4
**Correctness:** 3
**Technical Novelty And Significance:** 4
**Empirical Novelty And Significance:** 2
**Recommendation:** 5

**Clarity, Quality, Novelty And Reproducibility:**

The paper is clear and the application of diffusion models for topological ordering is novel.

The theory around uncovering the topological ordering appears fairly generic to me.  And I did not understand what aspects of it were novel.  Could the authors say more about how their theory relates to prior work? E.g. in what sense is Lemma 1 “more general” than Rolland 2022; I would have thought that the restriction to constant second-order derivatives would make the present result _less_ general by comparison.

**Strength And Weaknesses:**

Strenghts:
* The paper presents a timely analysis of how causal relationships may be uncovered using diffusion models.  Given that both causality and diffusion models are of interest to many in the community, the possible connections are certainly on the minds of many (including my own!).
*The paper runs with this direction and does a very nice job rigorously establishing some of what is possible with the Hessians of the log densities learned within diffusion probabilistic models.
*The paper is clearly written and the was enjoyable to read.

Weaknesses:
* Why not do the same analysis on a Gaussian kernel density estimate (KDE) of the data distribution? In my opinion, the paper is unpublishable without explicitly acknowledging this alternative route to the same ends, and how it would compare to the proposed method involving diffusion models.  In particular, in the limit of a large enough network / good enough optimization the Hessians computed coincide with what would obtain with the KDE.  So why go through the trouble of the diffusion model?  And is there anything special about diffusion models in this paper other than the fact that they provide computationally tractable Hessians (which is also possible with energy based and likelihood based models)?
* There may be good reasons to use a neural network here (e.g. amortization of Hessian computation, or the generalization properties may lead to better estimates of the Hessians of the population distribution as compared to the KDE). But these possibilities should be made explicit.  Without acknowledging that the approach involves what may be viewed as an approximation of something that may in principle be computed analytically, I view the paper as disingenuous.


**Summary Of The Paper:**

The structure of the Hessian of the data log likelihood carries information about topological ordering.  Since this information is learned by score based generative models, DPMs carry within them information necessary to reconstruct topological orderings.  The paper runs with this insight to obtain a practical algorithm with interesting scaling properties for causal discovery.

**Summary Of The Review:**

Please provide a short summary justifying your recommendation of the paper.
The paper rigorously explores a topic of interest to the community but does not comment on what would seem to me to be a more direct approach to solving the same problem (i.e. using a KDE).  The authors should comment on such alternatives to computing / approximating these Hessians that do not rely on diffusion models.

If the authors do not provide an explanation for why other approaches to computing the Hessian are ill-suited, or discuss how they might be applied here instead, I will significantly lower my score (to a reject recommendation).

---

> ### Author Response · Authors · 2022-11-11
> **Response to YaE5**
>
> We thank  the reviewer for the kind comments and thoughtful review, which are very constructive and helped us improve the draft. We are confident that our responses below will address you concerns.
>
> ### Q: Why diffusion models and how they compare with deep likelihood, KDEs or energy-based models?
>
> We agree with you, approximating the Hessians can be done with several techniques. We now include in the paper an extensive discussion acknowledging how this can be done and how we are different.
>
> We refer the reviewer to the “Why diffusion models” section of our “global response” for more details, where we discuss and compare DPMs with normalizing flows, VAE, KDEs and other energy based approaches. Also, we invite you to revisit the updated version which hopefully address your concerns about comparisons.
>
> In brief, KDEs are challenging when scaling to large datasets both in terms of memory and computing when evaluating the kernel matrices. This is the main motivation for exploring NNs in this context. Training NNs over large datasets give better approximations of the Hessian, therefore, better ordering results.
>
> ### Q: In what sense Lemma 1 is more general?
>
> The observation that the Hessian of the log-density of ANMs with GAUSSIAN noise is constant for leaf nodes comes from previous work (Rolland et al 2022). Indeed, constant second-order derivatives will only be true for Gaussian-like distributions (exponential with second-order polynomials) if only distributions supported over the whole real line are considered. Therefore, we make a note on the paper [end of Section 2.2] on how the two derivations are similar.

---

> > ### Comment · Reviewer_YaE5 · 2022-11-12
> > **Comparison to KDE is still insufficient**
> >
> > Regrettably, I feel my question / concern about why one might prefer the DPM approach to the KDE is insufficiently addressed by this response and the general comment.  For now I have lowered my score but hope the authors might follow up with further discussion or clarification (here and in the submission).
> >
> > My major concern is that is seems the Hessian of interest is more efficiently / easily computable or closely approximated in the settings the authors discuss than the authors let on.  I elaborate below.
> >
> > If I understand correctly, the estimand (with the KDE the DPM approximates) may be written as
> > $$
> > \nabla^2 \log p(x) = p(x)^{-1}H(x) + p(x)^{-2} g(x)g(x)^T,
> > $$
> > where $N$ is the number of datapoints, $p(x)=\frac{1}{N} \sum_{n=1}^N \mathcal{N}(x;\mu=x_n, \sigma^2 I),\ H(x)=\nabla_x^2 p(x)= \frac{1}{N} \sum_{n=1}^N \nabla_x^2 \mathcal{N}(x;\mu=x_n, \sigma^2 I),$ and $g(x)=\nabla_x p(x)=\frac{1}{N}\sum_{n=1}^N \nabla_x \mathcal{N}(x;\mu=x_n, \sigma^2 I)$
> > (see for example equation 8 [here](https://faculty.ucmerced.edu/mcarreira-perpinan/papers/cs-99-03.pdf) ).
> >
> > This is readily computed in $O(ND^2)$ time for any $x$ and may be trivially implemented with a few lines of python / numpy.  On my Macbook it takes <30 seconds for D=500 and N=10^5.  But even this full computation is not necessary as Monte Carlo approximations of $p(x), \ H(x),$ and $g(x)$ are very accurate with fewer than $O(N)$ samples (in fact I believe $\omega(\log D)$ suffices in the large $D$ limit).  And if one cares instead about $\sum_{n=1}^N \nabla^2 \log p(x_n)$ subsampling is again possible.
> >
> > Is it not the case that such an approach to Hessian computation is more straightforward (to implement and understand) as compared to training a diffusion model?
> >
> >
> > Re-lemma 1: thank you for this clarification.

---

> > > ### Author Response · Authors · 2022-11-14
> > > **Empirical comparison with KDEs and modelling clarifications**
> > >
> > >
> > > Thank you for clarifying your argument with examples and sharing the illuminating reference. We now better understand your comment regarding KDEs.
> > >
> > > Below, we provide additional clarification and discussion which we hope will convince to revise your score. Namely, we:
> > >
> > > 1. Clarify the setting and our modelling assumptions
> > > 2. Discuss the differences between the Stein estimator and the reviewer’s suggested estimator
> > > 3. Offer *numerical* evidence that approximating the Hessian with DPMs or Stein estimator directly is more suited for causal discovery.
> > >
> > > ### Clarifying Assumptions
> > >
> > > We start by clarifying our modelling assumptions, to elucidate the role of the DPM in calculating the score.
> > >
> > > In our setting, we assume Bayesian networks such that
> > > $$
> > > p(\mathbf{x}) = p(x_1, x_2, \cdots , x_d) = \prod_{i=1}^d p(x_i \mid Pa_i),
> > > $$
> > > where $d$ is the number of variables. We consider the relationship (also called a causal mechanism) between a variable $x_i$ and its parents $x_j \in Pa_i$ to take the form $x_i = f_i(Pa_i) + \epsilon_i$. If we isolate the deterministic from the stochastic part, and consider that each $\epsilon_i$ has a Gaussian distribution, we can re-write the likelihood as
> > > $$
> > > p(\mathbf{x}) = \prod_{i=1}^d \mathcal{N} \left( x,\mu = x_i - f_i(Pa_i), \sigma_i^2 \right).
> > > $$
> > >
> > > We do not know the functions $f_i(Pa_i)$ between variables, although we assume they are nonlinear. Since we do not have knowledge of these functions, we cannot derive the Hessian of the above log density analytically. Therefore, we choose a DPM to approximate the score of the Bayesian network.
> > >
> > > We note that the DPM is not approximating the score of a KDE as suggested in the comment “with the KDE the DPM approximates”, although both DPMs and KDEs are possible methods for approximation.
> > >
> > > ### Comparisons with KDE approximations
> > >
> > > [2] considered a second-order derivative of the KDE based on RBF kernels. Note, they use an extension of the Stein gradient estimator from [1], which provides a more accurate approximation of the Hessian than directly estimating $p(\mathbf{x})$, followed by computing the derivatives as a second step. The latter approach is described in [1] and is similar to what is proposed by the reviewer.
> > >
> > > **Intuition --** In detail, [1] show intuitively and empirically that the Stein estimator results in a more accurate approximation. Section 3.3.1 of [1] states that:
> > >
> > > > Inspecting and comparing it with the Stein gradient estimator, one might notice that the Stein method uses the full kernel matrix as the pre-conditioner, while the KDE method computes an averaged ``kernel similarity'' for the denominator. We conjecture that this difference is key to the superior performance of the Stein gradient estimator when compared to the KDE gradient estimator (see later experiments). [...] Thus it is reasonable to conjecture that the Stein method can be more sample efficient, which also implies higher accuracy when the same number of samples are collected.
> > >
> > >
> > > **Empirically--** To verify that the Stein gradient estimation is better suited for causal discovery than other KDE approaches, we ran some preliminary experiments with the KDE baseline suggested in [1] (implemented [here](https://github.com/YingzhenLi/SteinGrad/blob/master/began/helper_functions.py)) (KDE RBF), the suggested approach (KDE Gaussian), and compare with the Stein estimator used by [2] (Stein RBF). Indeed, the suggested KDE Gaussian is slighly is faster (because it does not require inverting a $n \times n$ matrix) but the performance is less accurate for causal discovery. Similarly to the paper, we report the order divergence (lower is better) of a dataset with $d = 20 $ nodes and $n = 1000$ samples. The results are:
> > >
> > > - KDE RBF – 40 (baseline inspired in [1]);
> > > - KDE Gaussian - 52 (suggested by reviewer).
> > > - Stein RBF – 10 (SCORE [2]);
> > > - [DPM – 9] (ours)
> > >
> > > These results suggest that computing the Hessian after estimation of $p(x)$ using the derived analytical equation is much less accurate than approximating the Hessian directly with the Stein estimator. These results also agree with the empirical results in [1].
> > >
> > > ### Conclusion
> > >
> > > We thank the reviewer for deepening the discussions around Hessian approximation approaches.  We will highlight in the paper that KDE approximations of the Hessian (such as the one referred to by the reviewer) are also possible.
> > >
> > > We would also like to highlight that the deciduous score update rule, derived in our paper, can potentially be extended to accelerate kernel-based estimation. This would allow the Hessian to be directly updated without the need to re-compute the kernel at each iteration. We leave this for future work.
> > >
> > > ### References
> > >
> > > [1] Yingzhen Li and Richard E Turner. Gradient Estimators for Implicit Models. ICLR 2018
> > >
> > > [2] Paul Rolland et al.  Score Matching Enables Causal Discovery of Nonlinear Additive Noise Models. ICML 2022

---

> > > > ### Author Response · Authors · 2022-11-18
> > > > **Response to YaE5**
> > > >
> > > > We hope we have addressed your concerns.

---

> > > > > ### Comment · Reviewer_YaE5 · 2022-11-19
> > > > > **Concerns partially addressed**
> > > > >
> > > > > I have updated my score.

---

### Official Review · Reviewer_1RL4 · 2022-10-25

**Confidence:** 4
**Correctness:** 2
**Technical Novelty And Significance:** 2
**Empirical Novelty And Significance:** 2
**Recommendation:** 5

**Clarity, Quality, Novelty And Reproducibility:**

- The paper overall is clear, while more justification is needed on their method, algorithm, and motivation.

- I mainly have concerns regarding novelty. As I commented above the proposed DiffAN method highly depends on the SCORE method proposed by Rolland et al., (2022). More clarity on the originality of the work is needed.

**Strength And Weaknesses:**

*Strength*

- The paper considers an important problem in causal discovery and leveraged a popular machine-learning tool.

*Weaknesses*

- The proposed DiffAN method highly depends on the SCORE method proposed by Rolland et al., (2022), by replacing the scoring part with diffusion probabilistic models (DPMs). More justification on novelty is needed. For instance, can the authors comment on if their method can be extended to other causal discoveries for non-linear models? In addition, please comment on why we cannot decouple DPMs from the SCORE method and consider a new causal discovery method built upon DPMs. I am also curious whether DPMs are really necessary for this approach. Why not consider other deep learning approaches since we are not handling images/texts?

- Although firstly introducing DPMs into causal discovery in this paper, the authors did not well explain why simply dropping the leaf in DPMs is sufficient for updating the score function without retraining. For instance, according to Algorithm 1, the scoring step is batch-dependent. If I understand correctly, when we update the score by pruning DPMs, it is not matching to the distribution of the next batch of data. In particular, the data $x$ is changing over batches in Equations 4 and 8. More theoretical justification is needed besides Figure 4.

- The numerical studies did not show the best performance of the proposed method. Indeed, Figure 3 suggests that both CAM and SCORE are better than the proposed DiffAN method. In addition, the runtime of SCORE in Figure 3 is still reasonable. Similarly, both GraN-DAG and SCORE perform better than the DiffAN method in real data  I would suggest running more studies that vote for the authors' method. Finally,
why not consider batches in SCORE so we can improve the run time as well?



**Summary Of The Paper:**

This paper considers non-linear causal discovery under the same setting as Rolland et al., (2022). The authors proposed to replace the scoring function in the SCORE method proposed by Rolland et al., (2022) with diffusion probabilistic models (DPMs). This allows them to update the learned Hessian without re-training the neural network and perform ordering over a batch, and thus they can large-scale data at a relatively faster speed. However, in total, the novelty of the approach and the profound impact are both limited. In addition, the main methodology and numerical results require additional justification.

**Summary Of The Review:**

Though the paper considers an important problem in causal discovery for the non-linear additive noise model, the novelty of the approach and the profound impact are both limited. In addition, the main methodology and numerical results require additional investigation.

---

> ### Author Response · Authors · 2022-11-11
> **Response to 1RL4**
>
> We thank the reviewer’s detailed and constructive comments. We appreciate that the reviewer agrees on the importance of the problem, which is commonly agreed upon across the reviewers.
>
> ### Q: “the proposed DiffAN method highly depends on the SCORE method proposed by Rolland et al., (2022).”
>
> Indeed, the observation that Var(Hessian) = 0 could be used for causal discovery on non-linear ANMs had been previously done in Rolland et al. (2022). Our method is highly based on their approach. However, our contribution is a mechanism (and theoretical results) for better scaling by replacing the kernel methods used in SCORE to high-dimensional and large datasets using NNs.
>
> ### Q: “The numerical studies did not show the best performance of the proposed method”
>
> Your comments encourage us to further investigate the scalability results. Please, we refer you to the global response for an in-depth discussion on experiments and the updated results.
>
> We would like to highlight that we don’t claim better performance for the results in Figure 3 nor the table with real data. Our main claim is around the scalability results. Indeed, we made an effort in the paper to clarify this point such as: (1) re-running the scalability experiments and showing better scaling; (2) moving the experiments with real data to the appendix. Indeed, very large real datasets as benchmarks for causal discovery are currently missing in the community. The current real datasets have less the $1000$ samples while we can scale up $10^5$.
>
> ### Q: “Can the authors comment on if their method can be extended to other causal discoveries for non-linear models?”, “Please comment on why we cannot decouple DPMs from the SCORE method and consider a new causal discovery method built upon DPMs.”
>
> There is prior art on neural causal discovery methods that rely on imposing constraints on the structure of the neural network architecture based on NOTEARS, for example. One could imagine loss functions that would optimise for score matching (via denoising diffusion) along with DAG constraints, instead of maximum likelihood. We haven’t explored this direction in our paper, but we believe this is an exciting direction which can definitely enjoy the scalability of DPMs!  We add this discussion to our conclusion.
>
> We couple DPMs with the SCORE (Rolland et al.) method, as a first attempt to explore the possibility of using DPMs for causal discovery. Our paper sheds light in how we can use DPMs to estimate the score  and subsequently efficiently estimate the Hessian for causal discovery. We agree that it might be possible to have a causal discovery method directly built upon DPMs (i.e. decoupling DPMs from the SCORE method). Yet, it is another hard but interesting problem to solve which we leave for future work as it deviates from the current focus of the paper.  The solution will have to be built from scratch in our opinion and definitely will be one of our future works.
>
> ### Q: “Why simply dropping the leaf in DPMs is sufficient for updating the score function without retraining.”
>
> A: We answer this question in Section 4.2, which states that “In addition, analysing Equation 5, the absolute value of the residue $\delta_l$ decreases if the values of $x_l$ are set to zero once the leaf node is discovered.”
>
> ### Q: “The scoring step is batch-dependent. When we update the score by pruning DPMs, it is not matching to the distribution of the next batch of data.”
>
> Not quite, the score at each step is computed over a subsample of the data. The main assumption is that the batch distribution (with a sufficiently large batch size, 512 in our case) approximates well the whole data distribution as assumed in many other prior works.
>
> ### Q: “why not consider batches in SCORE so we can improve the run time as well? ”
>
> Using the batches in kernel methods is not straightforward. Let us expand by adding the following explanation to the paper “Rolland et al.’s method cannot obtain gradient estimates at positions out of the training samples. Therefore, evaluating the Hessian over a subsample of the training dataset is not possible”. In particular, if we subsample the dataset used in SCORE, it’s the same as effectively using less data for training. This problem happens across most kernel methods. One potential solution could be using spectral decomposition [1]
>
> We hope that we have adequately addressed your concerns and would welcome any further feedback.
>
>
> [1] Jiaxin Shi et al. A Spectral Approach to Gradient Estimation for Implicit Distributions. ICML 2018

---

> > ### Comment · Reviewer_1RL4 · 2022-12-03
> > **Follow Up**
> >
> > I would like to thank the authors' effort on addressing my comments. Part of my concern has been resolved but I am not convinced by the assumption/approach, and the contribution still.
> >
> > In particular, the authors mentioned that *'The main assumption is that the batch distribution (with a sufficiently large batch size, 512 in our case) approximates well the whole data distribution as assumed in many other prior works'* in the response. I did not find this *main assumption* in the revised paper. Please correct me if I am wrong. In addition, may the authors clarify any specific *'other prior works'* here in the literature of causal discovery? My last and probably the largest concern regarding this point is why *'batch size, 512'* can *'approximate well the whole data distribution'* given the aim that the authors are handling high-dimensional and large datasets (say $n=10^5$)? I am sticking on this point as I suspect such a distributional shift causes bias in DPM training and thus leading to the current sub-optimal performance of the proposed method by the authors in the simulations in comparison to the original SCORE method.
> >
> > In addition, I appreciate the authors' great effort on elaborating why consider DPM is necessary for SCORE and show why it is better than deep learning and kernel method in the global response. But I think the authors did not address why we must consider the SCORE method as foundation of this work. This makes the current work less novel with less contribution, as the authors could conveniently leverage all the theorem and framework and just replace the kernel methods used in SCORE to DPM. I would like to see how we can utilize DPM in causal discovery, not DPM in SCORE.
> >
> > I have updated my score accordingly.

---

> > > ### Author Response · Authors · 2022-12-09
> > > **Response to follow up**
> > >
> > > We thank the reviewer for their reply and acknowledging our effort to clarify our assumptions and novelty.
> > >
> > > ### Q: “I suspect such a distributional shift causes bias in DPM training ”
> > >
> > > I would like to clarify that the DPM is trained over the entire train dataset. Therefore, training is not biased by subsampling. The batches in the present discussion are only explored at inference time. At inference time, the DPM has already been trained to approximate the Hessian. Batching helps accelerate the process such that the variance in eq. 2 of the paper is not computed over all datapoints.
> > >
> > > ### Q: ”why 'batch size, 512' can 'approximate well the whole data distribution' given the aim that the authors are handling high-dimensional and large datasets (say )?”
> > >
> > > Let us justify from another perspective (which we shall include in the final paper as well):
> > >
> > > When a node is a leaf, the corresponding Hessian’s diagonal is constant (as shown in Eq. 2). All other Hessian diagonal components vary, i.e. their variance is non-zero. The variance comes (if the DPM appropriately approximate the Hessian) from the derivatives of causal mechanisms w.r.t. their children. Intuitively, these should have non-zero variance even for very small data variations within a batch. We highlight that, at this point, the network is not learning anymore. Therefore, since the DPM is trained over the entire dataset, evaluation of the variance should be fairly robust to biases.
> > >
> > > ### Q: “I did not find this main assumption in the revised paper.”
> > >
> > > Indeed, we will make sure to clarify this assumption in the final version.
> > >
> > > ### Q: “why we must consider the SCORE method as foundation of this work”
> > >
> > > SCORE has two main contributions: (1) the Hessian’s diagonal for leaf nodes is constant; and (2) one can compute the Hessian via a second order Stein approximation. Yes, we utilise the former (1) as base for our work.
> > >
> > > However, the main argument is that using DPMs for computing the Hessian **is** valuable and novel in this settings. As described above, DPMs can be trained over very large datasets, proving better approximations of the Hessian, and therefore, better topological ordering.

---

### Official Review · Reviewer_F1bu · 2022-10-26

**Confidence:** 3
**Correctness:** 3
**Technical Novelty And Significance:** 3
**Empirical Novelty And Significance:** 3
**Recommendation:** 8

**Clarity, Quality, Novelty And Reproducibility:**

The structure of the article is good. There are some small issues, e.g.,   in definition 1, p(x)  should be a scalar, why p(x)  $\in R^{d}$?

**Strength And Weaknesses:**

Strength:
Integrating causal discovery and diffusion models is novel and interesting. The proposed method using diffusion methods to compute the Hessian of distribution scores is interesting.


Weakness:
The method only achieves comparable results on the data sets. The author should give detailed reasons why the method could not beat existing methods given that deep neural nets can provide better representations.

**Summary Of The Paper:**

The authors proposed DiffAN, a topological ordering algorithm that leverages DPMs for computing the Hessian. The algorithm updates the learned Hessian without re-training the neural network and performs ordering over a batch, which allows scaling to datasets with more samples and variables. In the experiments, they show that the method scales up well and achieves SOTA results.

**Summary Of The Review:**

The proposed method is novel and interesting.  The author should discuss the reason why the method cannot outperform the score-based method.

---

> ### Author Response · Authors · 2022-11-11
> **Response to F1bu**
>
> We thank the reviewer for their helpful comments. We are glad that the reviewer agrees on the importance of the problem, the novelty of the proposed method and the interesting aspects of the work, which are commonly agreed upon across the reviewers. Below, we answer two key questions raised.
>
> ### Q: “The method only achieves comparable results on the data sets”
>
> Your comment prompted us to further investigate our scaling experiments (our main empirical claim) and we found that increasing the NN capacity (making it deeper) considerably improved performance. Please, see global response for further details and Figure 4 for updated results in the submission.
>
> ### Q: Typo in Definition 1.
>
> We thank the reviewer for pointing this out. We carefully proof-read the draft and now fixed the issues mentioned in definition 1.

---

### Author Response · Authors · 2022-11-11
**Global Response**

We thank all reviewers for their comments. We are glad that reviewers overall agree on the importance and novelty of the proposed solution. In this global response, we would like to clarify our “main contribution” and answer two main questions common across reviews: “why diffusion models?” and “how results validate your contributions?”
We also updated the manuscript: Figure 4 with new results and textual changes are color-coded in blue.

### Clarifying contributions

The main result is that our method scales better compared to previous SoTA approaches for large causal graphs and with many observations. This is possible because the deciduous score allows NNs to be used without re-training!  Diffusion probabilistic models (DPMs), in particular, are a powerful tool for efficiently estimating the Hessians (see below).

We highlight that our theoretical results can potentially be used with other methods as long as they provide the score and its full Jacobian (not only the diagonals).  These are exciting directions for future work!

### Why diffusion models?

Diffusion probabilistic models (DPMs) are core to our models because they can efficiently learn the score and estimate its Jacobian. Thanks to the reviews, we have considerably expanded the comparison with previous work, to better position our work w.r.t. prior art. We also ran preliminary experiments. We invite the reviewers to re-visit the updated part in the “introduction”, “related works” and “conclusion” sections. (These are marked in blue in the updated submission for ease).

- DPMs or other deep models (scaling on $d$)

When comparing against other deep likelihood models such as normalising flows (NFs) or variational auto-encoders (VAEs), DPMs only require one backpropagation pass with grad(score) for obtaining the Hessian $O(d)$ as opposed to grad(grad(log p(x)) as in NFs $O(d^2)$, for example. Therefore, DPMs  scale better in number of variables / dimensionality of $d$.

Preliminary experiments with NFs (Neural Splines [2] and Real NVP [1]) showed that poor performance in high-dimensional (500 variables) settings. The algorithm for topological ordering with NFs would have complexity $O(n+d^4)$.

Compared to other energy-based models, DPMs are easy and stable to train (it’s only denoising) and scale to large dimensions. Deep models such as DKEF [3] that directly optimise the score-matching objective do not scale well to high-dimension because they need to compute the Hessian grad(grad(log p(x)) at each training step.

- DPMs or Kernels (scaling on $n$)

Our main baseline, SCORE (Rolland et al, 2022), uses a RBF kernel method. Computing the Hessian, however, required an extension of [4] based on Stein’s identity. In SCORE, the Hessian cannot be computed via backpropagation as in NNs. In addition, SCORE cannot obtain gradient estimates at positions out of the training samples – a known characteristic of kernel methods. Therefore, one cannot rely on subsampling (as we use in our paper) for evaluating the Hessian over a subsample of the training dataset. Most importantly, computing the kernel matrix is expensive for memory and computation when number of samples $n$ increase.

DPMs allow better scaling of number of variables $d$ than other likelihood deep models (which require two backprop passes) and better scaling in number of samples $n$ when compared to kernel methods.

### How results validate your contributions?

Our main focus is scalability for a variety of settings which we should highlight better.  In the submitted draft, we report our performance against baselines for small datasets to show that our method has similar accuracy in small-scale settings. Then, we show that our method scales to larger datasets where baselines are not tractable.

This question (thanks to the reviewer’s comments),   prompted us to further probe our scalability results (Figure 4, left) for bigger datasets. We re-ran the scalability experiments with deeper DNNs (MLP from 4 layers to 12 layers) and we found considerably increased performance. Effectively, we obtain better results than previous experiments with shallower networks.

We invite the reviewers to revisit Figure 4 (left) with the updated results!

We would also like to emphasise that we (as a community) need real data benchmarks with known GT graphs of large dimensionality.

---

> ### Author Response · Authors · 2022-11-11
> **References**
>
> [1] Laurent Dinh et al. Density estimation using Real NVP. ICLR, 2017.
>
> [2] Conor Durkan et al. Neural Spline Flows. NeurIPS, 2019.
>
> [3] Li  Wenliang et al. Learning deep kernels for exponential family densities. ICML, 2019
>
> [4] Yingzhen Li and Richard E Turner. Gradient Estimators for Implicit Models.  ICLR, 2018.

---

> ### Comment · Reviewer_YaE5 · 2022-11-12
> **DPM Hessian computation in O(D)?**
>
> Thank you for your thoughtful rebuttal.
>
> Can you elaborate on the time complexity of Hessian computation with DPMs?  If I understand correctly, the claim is computing the Hessian (which as D^2 entries) requires only O(D) time.  Writing D^2 entries into memory should be O(D^2) time.  Is this a typo or am I missing something else?

---

> > ### Author Response · Authors · 2022-11-14
> > **DPM Hessian requires $O(D)$**
> >
> > Yes, we endeavour to elaborate here.
> >
> > Computing the Hessian requires only $O(D)$ with diffusion models because each backpropagation pass computes an entire row of the Hessian at once.
> >
> > This is because for the diffusion model $\epsilon_\theta \in R^{D+1} \rightarrow R^D$, in which $\epsilon_\theta$ approximates $\nabla \log p(\mathbf{x})$, computing the Hessian requires deriving each of the $D$ outputs with respect to each of the $D$ inputs. However, each row of the Hessian $\mathbf{H}_{i,:}$ is computed in parallel for one backward pass of backpropagation, therefore, the procedure only needs to be performed D times.

---

### Decision · Program_Chairs · 2023-01-20

**Decision:**

Accept: poster

**Justification For Why Not Higher Score:**

A good paper that improves state of the art and improves the scalability of causal discovery methods.

**Justification For Why Not Lower Score:**

N/A

**Metareview: Summary, Strengths And Weaknesses:**

The paper introduced a new approach for finding a topological ordering based on identifying low variance entries along the diagonal of the Hessian of the log density. Efficiency is obtained, as opposed to previous methods, by not having to reestimate the entire Hessian to identify each leaf node, which is implemented using diffusion models. The results show that the new method is comparable to the previous state-of-the-art but is more efficient in high dimensions.

The reviewers had some mixed feelings about the paper, in particular about its clarity. As mentioned by reviewer vVJW, while the work is described as "diffusion models for causal discovery," diffusion models are only leveraged for their estimates of the score function (as opposed to their generative modeling capabilities). This means that any nonparametric density estimation approach (with backprop) could be used instead of the diffusion model. I think the discussion could be more interesting if these things were disentangled, which I leave as a suggestion for improvement to the authors. Also, reflect the reviewers' comments and discussion, including vVJW, in the updated version of the manuscript.

**Note From Pc:**

if the above contains the word "oral" or "spotlight" please see: "oral" presentation means -> notable-top-5% and "spotlight" means -> notable-top-25%. As stated in our emails, we are disassociating presentation type from AC recommendations